

# Within-person structures of daily cognitive performance differ from between-person structures of cognitive abilities

Florian Schmiedek[1,2], Martin Lövdén[1,3], Timo von Oertzen[1,4] and Ulman Lindenberger[1,5,6]

[1] Center for Lifespan Development, Max Planck Institute for Human Development, Berlin, Germany
[2] DIPF | Leibniz Institute for Research and Information in Education, Frankfurt am Main, Germany
[3] Aging Research Center, Karolinska Institutet, Stockholm, Sweden
[4] Current affiliation: Department of Psychology, Universität der Bundeswehr München, München, Germany
[5] European University Institute, San Domenico di Fiesole, Italy
[6] Max Planck UCL Centre for Computational Psychiatry and Ageing Research, London & Berlin, United Kingdom & Germany

## ABSTRACT

Over a century of research on between-person differences has resulted in the consensus that human cognitive abilities are hierarchically organized, with a general factor, termed general intelligence or "*g*," uppermost. Surprisingly, it is unknown whether this body of evidence is informative about how cognition is structured within individuals. Using data from 101 young adults performing nine cognitive tasks on 100 occasions distributed over six months, we find that the structures of individuals' cognitive abilities vary among each other, and deviate greatly from the modal between-person structure. Working memory contributes the largest share of common variance to both between- and within-person structures, but the *g* factor is much less prominent within than between persons. We conclude that between-person structures of cognitive abilities cannot serve as a surrogate for within-person structures. To reveal the development and organization of human intelligence, individuals need to be studied over time.

## INTRODUCTION

The quantitative measurement of intelligence is one of the greatest accomplishments in the behavioral sciences (*Nisbett et al., 2012*). A century or more of research has resulted in a consensus view that human cognitive abilities are hierarchically organized (*Carroll, 1993*). At the bottom of the hierarchy, numerous specific abilities, such as numerical reasoning or verbal fluency, can be identified. Differences between individuals in specific abilities form broader abilities like reasoning or episodic memory, which again show substantial positive correlations with one another. This pattern has led researchers to postulate the concept of a general cognitive ability, or "*g*," at the top of the hierarchy (*Jensen, 1998*; *Spearman,*

Corresponding author
Florian Schmiedek,
schmiedek@dipf.de

*1927*). Often equated with the term "intelligence," the *g* factor is a dominant predictor of between-person differences in real-life outcomes such as educational success, vocational achievement, health, and mortality (*Batty, Deary & Gottfredson, 2007*; *Deary et al., 2007*; *Gottfredson & Deary, 2004*; *Schmidt & Hunter, 1998*; *Strenze, 2007*).

Virtually all of the evidence on the hierarchical structure of human intelligence is based on associations among between-person differences in performance on batteries of cognitive tasks. A large body of research shows that both genetic and epigenetic differences (e.g., reflecting birth weight, nutrition, formal schooling, etc.) contribute to the hierarchical organization of these between-person differences (*Deary, 2001*). However, it is likely that many factors contributing to differences *between* individuals vary less, or differently, *within* individuals. Examples are allelic variations of the genome, which are present between but not within individuals. Conversely, the factors that contribute to variations within persons over time may contribute little to average between-person differences. The effects of weather conditions on cognitive performance may be an example—at least for people living in the same place. Besides these pronounced examples, there is a host of factors that may influence both differences between persons as well as variation within persons over time. For example, people differ from each other in their average level of motivation and they vary in their momentary levels of motivation over time (*Brose et al., 2010*). These different factors may potentially influence all tasks (contributing to the *g* factor), only tasks of one or more of the broader or narrower abilities (contributing to the variance of the corresponding ability factors), or only single tasks (contributing to the variance of just the corresponding task), and they might do so to different degrees at the different levels of analysis. Furthermore, the different factors that are operating might be correlated to different degrees across persons and/or across time. It can therefore be expected that corresponding correlation structures at the between-person and the within-person level could only be found after accounting for all the factors that differentially affect the different levels (*Voelkle et al., 2014*).

Without taking into account these factors, many of which are probably unknown or unobservable, there is no strong theoretical reason to expect a close correspondence between within-person and between-person structures of cognitive abilities. In technical terms, within-person and between-person structures can be expected to be non-ergodic (*Molenaar, Huizenga & Nesselroade, 2003*). Ergodicity describes the equivalence of between-person and within-person structures and requires the homogeneity of within-person structures (i.e., individual within-person structures not differing from each other) as well as stationarity of the means and (co)variances (i.e., no changes of distributions across time). As an illustration, imagine that episodic memory and working memory correlate $r = .70$ when assessed in 100 different individuals at a single occasion. Further consider that each of these 100 individuals is assessed on 100 different days on the same two sets of measures, and correlations are computed for each individual separately across the 100 days. How much do within-person correlations of these 100 individuals differ from each other? Will an observed between-person correlation of $r = .70$ fall within or outside the distributional range of the 100 within-person correlations?

While considerable differences of within-person correlations across individuals, as well as from between-person correlations, have been demonstrated for the domain of affect (*Brose et al., 2015*; *Fisher, Medaglia & Jeronimus, 2018*), these questions await empirical testing in the domain of cognition. Though ergodicity is unlikely to hold in that domain, the structure of between-person variation has often been treated as a proxy or surrogate for the organization of intelligent behavior at the individual level, both in psychology and in cognitive neuroscience. This research practice has become subject to challenge on theoretical grounds, necessitating a direct formal comparison of between-person and within-person structures of cognitive functioning (*Borsboom, Mellenbergh & Van Heerden, 2003*; *Kievit et al., 2013*; *Lautrey, 2003*; *Molenaar, 2004*). However, to the best of our knowledge, no comprehensive investigation of the correspondence between within- and between-person structures of cognitive abilities has been reported thus far. Given that repeated assessment of cognitive performance typically leads to changes in individuals' levels of performance (e.g., practice-related improvements, motivation-related performance decrements), expecting ergodicity is unrealistic from the outset. Yet, an empirical investigation and quantification of the degree of non-correspondence of between- and within-person structures is required to lay the foundation for further research on the (potential manifold of) mechanisms that explain such non-correspondence.

Hence, we conducted the COGITO study, in which 101 adults aged 20 to 31 years worked on a battery of nine cognitive tasks (plus three tasks not included in the present investigation) on over 100 daily occasions. In an earlier report, we demonstrated the presence of reliable day-to-day fluctuations in cognitive performance within individuals (*Schmiedek, Lövdén & Lindenberger, 2013*; for similar results, see *Rabbitt et al., 2001*). For each of the nine tasks, we found reliable performance differences across days, beyond the degree of variation that can be attributed to within days (i.e., across blocks of trials within a testing session). Here, we determine the degree of similarity between within-person and between-person structures of cognitive abilities using the Kullback–Leibler (KL) divergence (*Kullback & Leibler, 1951*). Specifically, we investigate whether correlation structures based on between-person differences are similar to within-person structures based on repeated daily assessments. The KL divergence is an appropriate metric for this question because it provides a symmetrical measure of how much information (measured in nats = 1.44 bits) is lost when one statistical distribution (i.e., a between-person correlation matrix) is used to describe another distribution (i.e., a within-person correlation matrix; see below for further information). See Supplemental Material 1 for detailed information on why the KL divergence is an appropriate measure to describe the difference between correlation matrices.

## MATERIALS & METHODS

### Participants and procedure

During the daily assessment phase of the COGITO Study, 101 younger adults (52 women, age: 20–31 years, $M = 25.6$, $SD = 2.7$) completed an average of 101 practice sessions. Participants with a target age from 20 to 30 years were recruited through newspaper

advertisements, word-of-mouth recommendation, flyers distributed in university buildings, community organizations, and local stores. The advertisements were aimed at people interested in practicing cognitive tasks for 4–6 days a week for a period of about 6 months. Several steps were required for inclusion in the study. First, interested people were given information about the study in a telephone interview and checks were made as to whether the requirements for participation in the study, in particular the time investment, could be met. There were no further exclusion criteria regarding participant characteristics. Potential candidates for participation were then called back and invited to join a 1-hour "warm-up" group session in which general aims of the study were explained and detailed information on incentives was given. At the end of this session, individuals were able to decide to take part in the study. The sample was quite representative regarding general cognitive functioning, as indicated by comparisons of Digit-Symbol performance with data from a meta-analysis (*Schmiedek, Lövdén & Lindenberger, 2010*). The attrition rate for those participants who had entered the longitudinal practice phase was low (i.e., of 120 participants who completed the pretest sessions, 19 dropped out from the study before finishing the 100 daily sessions; for further details on dropout rates and reasons for dropout in the different study phases, see (*Schmiedek et al., 2010*). On average, participants had 12.5 years of high school education; 98% were single and 2% were married; the majority were university students (67%); the remainder were employed (13%), unemployed (11%), school students (8%), or apprentices (1%). Participants practiced individually in lab rooms containing up to six computer testing places. They could come to the lab and do testing sessions on up to six days per week (Mondays to Saturdays). On average, it took participants 197 days to complete the 100 sessions. Before and after this longitudinal phase, participants completed pre- and posttests in ten sessions that consisted of 2–2.5 h of comprehensive cognitive test batteries and self-report questionnaires. Participants were paid between 1450 and 1950 Euros, depending on the number and temporal density of completed sessions. The ethical review board of the Max Planck Institute for Human Development, Berlin, approved the study. All research was performed in accordance with relevant guidelines. Informed written consent was obtained from all participants.

## Tasks

In each practice session, participants practiced twelve different tasks drawn from a facet structure cross-classifying cognitive abilities (perceptual speed, episodic memory, and working memory) and content material (verbal, numerical, figural-spatial) with two to eight blocks of trials each (for information on all practiced tasks, see *Schmiedek, Lövdén & Lindenberger, 2010*). Three of a total of six tasks of perceptual speed were choice reaction tasks that were included to measure basic aspects of information processing. They were not considered in the current analyses. Here, we used three comparison tasks of perceptual speed that are more typical for cognitive test batteries applied in research on the structure of intelligence (see below for information on tasks applied here).

For the episodic and working memory tasks, presentation time (PT) was adjusted individually based on pretest performance. For each task and each individual, mean

accuracies for the different PT conditions at pretest were fitted with exponential time-accuracy functions (including freely estimated parameters for onset, rate, and asymptote as well as a lower asymptote parameter fixed to different values for each task, which was .10 for memory updating, .50 for 3-back, and .00 for Alpha Span and the episodic memory tasks). The fitted values from these functions were used to choose PTs that are clearly above random guessing but below some upper level. The upper level was defined by the midpoint between the lower asymptote level and perfect accuracy [e.g., $(0.10 + 1.0)/2 = 0.55$ for Memory Updating; see below], while the minimum level was defined by the midpoint between the lower asymptote level and the upper level [e.g., $(0.10 + 0.55)/2 = 0.325$ for Memory Updating]. If performance was above the upper level for the second-fastest PT, the fastest PT was chosen even if predicted accuracy was below the minimum level for the fastest PT. For the Alpha Span task, we deviated from the described procedure and chose 0.40 as the minimum level and 0.60 as the upper level on the basis of empirically observed time-accuracy functions.

### Perceptual speed: comparison tasks

In the numerical, verbal, and figural versions of the comparison task, either two strings of five numbers or digits each, or two colored three-dimensional objects consisting of several connected parts ("fribbles") appeared on the left and right portion of the screen. Participants had to decide as quickly as possible whether both stimuli were exactly the same or different. If different, the strings differed only by one number or letter and the objects differed only by one part. Number strings were randomly assembled using digits 1 to 9. Letters were lower case and randomly assembled from all consonants in the alphabet, thus ensuring that they could not actually form real words. In each session, two blocks of 40 items were included with equal numbers of same and different stimuli. Images of fribbles used in this task are courtesy of Michael J. Tarr, Brown University, http://www.tarrlab.org/.

All three comparison tasks were scored by dividing the number of correct responses by the total response time (in seconds) and multiplying this quotient by 60 (i.e., creating a score of correct responses per minute).

### Episodic memory tasks

- *Verbal episodic memory: Word Lists.* Lists of 36 nouns were presented sequentially with PTs of 1000, 2000, or 4000 ms, and an interstimulus interval (ISI) of 1000 ms. Word lists were assembled so as to balance word frequency, word length, emotional valence, and imageability across lists. After presentation, words had to be recalled in correct order by entering the first three letters of each word using the keyboard. Two blocks were included in each daily session. The performance measure was based on the percentage of correctly recalled words multiplied by a score ranging from 0 to 1, which represented the correctness of the order (based on a linearly rescaled tau rank correlation). The resulting scores were logit-transformed before entering the analyses.
- *Numerical episodic memory: Number-Noun Pairs.* Lists of 12 two-digit numbers and nouns in plural case pairs were presented sequentially with PTs of 1000, 2000, or 4000 ms; and an ISI of 1000 ms. After presentation, all numbers had to be entered based on

random noun prompts. Two blocks were included in each daily session. The performance measure used in the analyses was the logit-transformed percentage of number of correctly recalled numbers.

- *Figural-spatial episodic memory: Object Position Memory.* Sequences of 12 coloured photographs of real-world objects were displayed at different locations in a six-by-six grid with PTs of 1000, 2000, or 4000 ms, and an ISI of 1000 ms. After presentation, objects appeared at the bottom of the screen and had to be moved to the correct locations in the correct order by clicking on objects and locations with the computer mouse. Two blocks were included in each daily session. The performance measure was the percentage of items placed in the correct locations multiplied by a score ranging from 0 to 1, which represented the correctness of the order (based on a linearly rescaled tau rank correlation). The resulting scores were logit-transformed before entering the analyses.

## Working memory tasks

- *Verbal working memory: Alpha Span.* Ten upper-case consonants were presented sequentially together with a number located below the letter. For each letter, participants had to decide as quickly as possible whether the number corresponded to the alphabetic position of the current letter within the set of letters presented up to this step. Five of the ten items were targets. If position numbers were incorrect (non-targets), they differed from the correct position by +/- one. PTs were 750, 1500, or 3000 ms, and the ISI was 500 ms. Eight blocks were included in each daily session. The performance measure used in the analyses was based on the percentages of correct responses. Scores were averaged across odd and even blocks and logit-transformed.

- *Numerical working memory: Memory Updating.* Participants had to memorize and update four one-digit numbers. In each of four horizontally placed cells, one of four single digits (from 0 to 9) was presented simultaneously for 4000 ms. After an ISI of 500 ms, a sequence of eight "updating" operations were presented in a second row of four cells below the first one. The updating operations were subtractions and additions from −8 to +8. The updating operations had to be applied to the digits memorized from the corresponding cells above and the new results then also had to be memorized. Each updating operation was applied to a cell different from the preceding one, so that no two updating operations had to be applied to one cell in sequence. PTs were 500, 1250, or 2750 ms, and the ISI was 250 ms. The final result for each of the four cells had to be entered at the end of each trial. Eight blocks were included in each daily session. The performance measure used in the analyses was based on the percentages of correct responses. Scores were averaged across odd and even blocks and logit-transformed.

- *Spatial working memory: 3-Back.* A sequence of 39 black dots appeared at varying locations in a four-by-four grid. For each dot, participants had to determine whether it was in the same position as the dot three steps earlier in the sequence or not. Dots appeared at random locations with the constraints that (a) 12 items were targets, (b) dots did not appear in the same location at consecutive steps, (c) exactly three items each were 2-, 4-, 5-, or 6-back lures, that is, items that appeared in the same position as they had 2, 4, 5, or 6 steps earlier. The presentation rate for the dots was individually

adjusted by varying ISIs (500, 1500, or 2500 ms). PT was fixed at 500 ms. Four blocks were included in each daily session. The performance measure used in the analyses was based on the percentages of correct responses on trials 4-39. Scores were averaged across odd and even blocks and logit-transformed.

## Validity of the tasks

To evaluate the validity of our tasks for the assessment of cognitive abilities, we made use of an established paper-and-pencil intelligence test battery, the Berlin Intelligence Structure (BIS) Test (*Jäger, Süß & Beauducel, 1997*), which included the cognitive ability factors of perceptual speed, episodic memory, and reasoning (used here as the criterion ability for working memory).

For the perceptual speed tasks, the latent correlation with the corresponding BIS factor at pretest was .58, while the correlations with reasoning and episodic memory in the BIS were .25. At posttest, the correlation with perceptual speed in the BIS significantly decreased to .28, whereas the correlations to reasoning and episodic memory did not change significantly (Table 1). For the working memory tasks, the latent correlations with reasoning ranged from .82 to .96 at pretest (for the different presentation times), and decreased to .50–.68 at posttest, with differences being significant for the two slower presentation time conditions. The correlations with perceptual speed and episodic memory in the BIS did not differ significantly between pretest and posttest (Table 2). For our EM tasks at pretest, the latent correlations with the BIS episodic memory factor ranged from .76 to .82 and were lower for reasoning (.51–.54) and for perceptual speed (.51–.52). At posttest, none of the correlations differed significantly from the correlations at pretest (Table 3). In sum, in line with early suggestions (*Hofland, Willis & Baltes, 1981*; *Labouvie et al., 1973*), there were some indications that perceptual speed and working memory lost some of their criterion validity in the course of task exposure, when taking paper-and-pencil based assessments as reference. Because of this, we included the posttest scores into the comparisons of between-person and within-person structures.

## Data analysis
### De-trending

To investigate how strongly the divergence of between-person and within-person structures was driven by practice-related changes in performance levels (i.e., trends), all analyses were carried out with raw data and de-trended data. The de-trended data were computed by first smoothing every within-person time series using a Gaussian filter with a standard deviation of three sessions. Afterwards, the smoothed time series was subtracted from the raw time series to obtain the de-trended time series. The algorithm used is part of the Onyx SEM software system backend (*von Oertzen, Brandmaier & Tsang, 2015*).

### Kullback–Leibler divergences

Distances between correlation structures were computed as the symmetrical KL divergence (*Kullback & Leibler, 1951*). The KL divergence of two distributions A and B is the number of information units lost when describing a random variable by A if it follows B. The symmetrical KL divergence is the sum of the distance from A to B and the distance from

**Table 1  Correlations of the perceptual speed factor to ability factors of the Berlin Intelligence Structure Test.**

| BIS-PS | |
|---|---|
| Pretest | **.578** |
| Posttest | **.278** |
| $\chi^2$ Test of Difference | 11.275 |
| **BIS-Reasoning** | |
| Pretest | .245 |
| Posttest | .146 |
| $\chi^2$ Test of Difference | 0.756 |
| **BIS-EM** | |
| Pretest | .252 |
| Posttest | .159 |
| $\chi^2$ Test of Difference | 0.944 |

Notes.

Differences between pretest and posttest correlations were tested with likelihood-ratio tests, comparing the model in which the correlation were freely estimated with a model in which it was constrained to be equal. The resulting $\chi^2$ tests all have $df = 1$ and a critical value (with $\alpha = .05$) of 3.841; significant differences (pretest vs. posttest) are shown in bold face.
BIS, Berlin Intelligence Structure Test; PS, Perceptual Speed; EM, Episodic Memory.

**Table 2  Correlations of the working memory factor to ability factors of the Berlin Intelligence Structure Test.**

| | Presentation Time Condition | | |
|---|---|---|---|
| | 1 | 2 | 3 |
| **BIS-PS** | | | |
| Pretest | .703 | .639 | .609 |
| Posttest | .500 | .433 | .394 |
| $\chi^2$ Test of Difference | 1.569 | 3.188 | 3.175 |
| **BIS-Reasoning** | | | |
| Pretest | .819 | **.957** | **.868** |
| Posttest | .679 | **.505** | **.500** |
| $\chi^2$ Test of Difference | 0.938 | 20.699 | 13.691 |
| **BIS-EM** | | | |
| Pretest | .505 | .680 | .615 |
| Posttest | .683 | .515 | .624 |
| $\chi^2$ Test of Difference | 1.186 | 2.337 | 0.007 |

Notes.

Differences between pretest and posttest correlations were tested with likelihood-ratio tests, comparing the a model in which the correlation were freely estimated with a model in which it was constrained to be equal. The resulting $\chi^2$ tests all have $df = 1$ and a critical value (with $\alpha = .05$) of 3.841; significant differences (pretest vs. posttest) are shown in bold face.
BIS, Berlin Intelligence Structure Test; PS, Perceptual Speed; EM, Episodic Memory.

B to A. For normal distributions with covariance matrices $\Sigma_1$ and $\Sigma_2$ of $K$ variables, the symmetrical KL is given by

$$symKL\ (\Sigma_1,\ \Sigma_2) = 2K + Tr(\ \Sigma_1 \Sigma_2^{-1} + \Sigma_2\ \Sigma_1^{-1}).$$
**Table 3 Correlations of the episodic memory factor to ability factors of the Berlin Intelligence Structure Test.**

|  | Presentation time condition | | |
|---|---|---|---|
|  | **1** | **2** | **3** |
| **BIS-PS** | | | |
| Pretest | .517 | .507 | .516 |
| Posttest | .405 | .407 | .426 |
| $\chi^2$ Test of Difference | 2.205 | 2.162 | 1.983 |
| **BIS-Reasoning** | | | |
| Pretest | .509 | .506 | .543 |
| Posttest | .489 | .416 | .443 |
| $\chi^2$ Test of Difference | 0.066 | 1.693 | 2.392 |
| **BIS-EM** | | | |
| Pretest | .822 | .759 | .790 |
| Posttest | .698 | .677 | .708 |
| $\chi^2$ Test of Difference | 3.449 | 1.785 | 1.859 |

**Notes.**

Differences between pretest and posttest correlations were tested with likelihood-ratio tests, comparing the a model in which the correlation were freely estimated with a model in which it was constrained to be equal. The resulting $\chi^2$ tests all have $df = 1$ and a critical value (with $\alpha = .05$) of 3.841; significant differences (pretest vs. posttest) are shown in bold face.

BIS, Berlin Intelligence Structure Test; PS, Perceptual Speed; EM, Episodic Memory.

## Statistical testing with KL divergences

To establish that the differences of the within-person correlation matrices from each other and from the between-person centroid are significant, a null distribution was sampled, and the actual divergences were compared to this distribution. We simulated the same data structure as that of the actual data, namely, 101 data lines with nine tasks, under the null hypothesis that the underlying correlation matrix is the same for all participants, that is, either the within-person centroid or the between-person centroid. The average symmetrical KL divergence in the simulated data was computed for each of 10,000 trials, either the KL divergence of all within-person pairs or the distance from each within-person pair to the between-person centroid, respectively. The actual average symmetrical KL divergence was then compared to this distribution. If, for example, the actual average symmetrical KL divergence is within the highest 5% difference of the simulated trials, this indicates a significant rejection of the null hypothesis with $\alpha = 5\%$.

## Confidence Intervals for KL divergences

To further demonstrate the difference of the within-person correlation matrices from the between-person centroid, we also sampled the distribution of KL divergences for every within-person correlation matrix. For each participant, we again simulated, in 10,000 trials, 101 data lines with nine tasks under the assumption that the descriptive within-person correlation matrix was the true correlation matrix. This provided us with a distribution of KL divergences between the between-person centroid and the distribution of correlation matrices for each participant, and in turn with a confidence interval for the KL-divergence for every individual participant.

### Multidimensional scaling (MDS)

To illustrate the distance between within-person and between-person correlation matrices, KL divergences were embedded in a lower-dimensional space that preserves the maximal precision of the pairwise differences using MDS (*Torgerson, 1958*). MDS finds a vector of coordinates for every correlation matrix such that the Euclidean distances between pairs of vectors are closest to the KL distances of the correlation matrices. A property of MDS is that a solution for fewer dimensions is a projection from the solutions for more dimensions, that is, the coordinates of the first dimensions are always the same for any number of dimensions in the MDS. A plot of the first two coordinates is read as an illustration of the distances of the covariance matrices. The MDS was computed using an algorithm that is part of the Onyx SEM software system backend (*von Oertzen, Brandmaier & Tsang, 2015*).

### Hierarchical factor models of centroid correlation matrices

Centroid correlation matrices based on the between-person and the raw or de-trended within-person data were calculated as the component-wise average of all correlation matrices. These correlation matrices were then submitted to confirmatory factor models (using SAS PROC CALIS) imposing a hierarchical structure, with tasks loading on three ability factors (i.e., perceptual speed, working memory, and episodic memory) that in turn loaded on a general factor (thereby forming a saturated second-order factor sub-model).

## RESULTS

For the present analyses, we used nine cognitive tasks that are (a) suitable for intensively repeated assessments and (b) representative of broad ability factors in established hierarchical models of intelligence. Specifically, the tasks represent perceptual speed with comparison tasks, episodic memory with different recall tasks, and different working memory paradigms. The latter were chosen because of the close relation of working memory to the important factor of fluid intelligence/reasoning in our study (*Schmiedek, Lövdén & Lindenberger, 2014*) and in the literature (*Conway, Kane & Engle, 2003*; *Duncan, 2013*; *Kyllonen & Christal, 1990*; *Wilhelm, Hildebrandt & Oberauer, 2013*), and the fact that they are much better suited for repeated assessment across 100 occasions than typical reasoning tasks. Latent factor correlations with ability factors from an established paper-and-pencil test of intelligence showed that the ability factors of the practiced tasks show patterns of good convergent and discriminant validity at pretest, which do shift to some degree at posttest (see Method: Validity of the tasks, for details). Presentation times of episodic memory and working memory tasks were individually adjusted based on pretest performance to avoid floor or ceiling effects, and then kept constant throughout the daily testing occasions. At pretest and posttest, participants worked on all tasks under all possible presentation time conditions, providing reliable measurements of between-person correlation structures that correspond to each of the presentation time constellations of the within-person covariance structures. That is, for each individual pattern of presentation time conditions of the 101 participants, the corresponding presentation time conditions from the pretest (or posttest) data could be picked to compute a between-person correlation matrix that matches the presentation times of this participant's within-person data. As the correlations with the

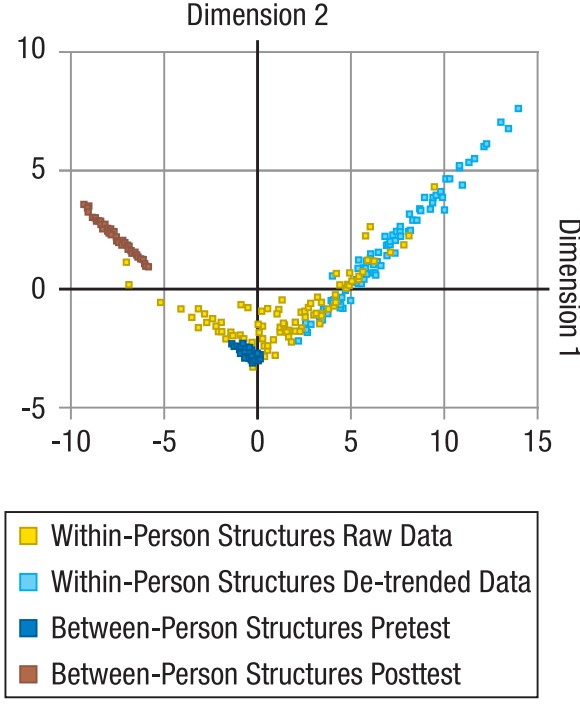

**Figure 1** **Comparisons of between-person and within-person structures of cognitive abilities.** Locations of within-person (raw data: yellow dots; de-trended data with longer-term trends taken out: light blue dots) and between-person structures (at pretest: dark blue dots; at posttest: brown dots) on the first two dimensions of a multidimensional scaling solution for the Kullback-Leibler (KL) divergences between all within- and between-person structures. Between-person structures are based on performance of the same sample on the same tasks under different presentation time conditions, and are relatively similar to each other. Within-person structures evidently differ more from each other and clearly overlap little (for raw data) or nor not at all (for de-trended data) with the between-person structures.

abilities of the paper-and-pencil intelligence test did change from pretest to posttest, we included both the between-person structures from pretest and from posttest into the analysis to be able to evaluate the between/within differences in relation to the changes of the between-person structures from pretest to posttest.

For all unique comparisons of the resulting 202 between-person (101 from pretest and 101 from posttest) and 202 within-person correlation matrices (101 based on raw data and 101 based on de-trended data), a total of 163,216 ($= 404*404$) KL divergences were calculated. These distance measures were then submitted to MDS to represent the relative distance of the within-person matrices to the between-person matrices, and of the within-person matrices (or between-person matrices) to each other in a low-dimensional space (Fig. 1).

We found that within-person structures based on raw data differed reliably from the corresponding between-person structures from pretest (average KL divergence $= 5.90$; $p < .001$; for information on how $p$ values were determined, see Data analysis: Statistical testing with KL divergences), and among each other (average KL divergence $= 6.84$; $p < .001$; $SD_{Dimension1} = 3.66$; $SD_{Dimension2} = 1.81$).

When simulating data from each individual within-person correlation matrix, 95% Confidence Intervals for the KL divergence to the between-person centroid ranged from [1.93; 4.03] for the person closest to the between-person centroid to [12.24; 24.21] for the person furthest away from the between-person centroid. Importantly, the confidence intervals of the KL divergences of the within-person correlation matrices to the between-person centroids in no case included 0.85, the average KL divergence we would expect when the within-person data were generated from the between-person correlation matrix.

To illustrate what an average KL divergence of 5.90 means, it may be helpful to compare this value to one that would result from a comparison of correlation matrices that differ in a theoretically meaningful way. Consider three correlation matrices, (1) one that represents the average between-person correlation matrix from our pretest data, which can be modelled by a hierarchical factor model with three first-order factors and a general factor on top, (2) a model with the same correlations among the tasks that belong to one ability, but zero correlations between tasks from different abilities, and (3) a model corresponding to (2), but with the factor correlations fixed to one. The second and third models would therefore represent orthogonal factors (akin to a Thurstonian model of independent abilities) versus a single-factor model (akin to a Spearmanian model of a single g factor). The resulting KL divergence between the observed between-person correlation matrix and the Thurstonian model (2) was 0.80, while the divergence between the observed between-person correlation matrix and the Spearmanian model (3) was 11.36. The average divergence of the within-person correlation matrices from the between-person correlation matrices was therefore right in between these two theoretically meaningful and substantial differences.

To further illustrate a KL divergence of 5.90, we picked the participant with an individual KL divergence (of 5.95) closest to this average value. Comparing the within-person and corresponding between-person structures of this participant (see Table S1 clearly shows sizeable and systematic differences. For example, within-person correlations between tasks of perceptual speed and tasks of working memory are higher than corresponding between-person correlations, while the within-person correlations among tasks of episodic memory were lower.

When within-person data were first de-trended to account for longer-term trends such as practice-related improvements (for details, see Data analysis: De-trending), within- and between-person structures from pretest showed no overlap at all (Fig. 1; difference between within- and between-person structures from pretest: average KL divergence = 5.67; $p < .001$; differences among within-person structures for de-trended data: average KL divergence = 3.01; $p < .001$; $SD_{Dimension\ 1} = 2.57$; $SD_{Dimension\ 2} = 2.14$). For raw data, MDS Dimension 1 (horizontal) correlated strongly with the magnitude of the first eigenvalue of the within-person correlation structures ($r = -.78$; $p < .001$). For de-trended data, MDS Dimension 1 fully separated all within- from all between-person structures and was again strongly correlated with the first eigenvalue of the within-person structures ($r = -.59$; $p < .001$). Together, this indicates that the size of the differences between within- and between-person structures was associated with the degree to which longer-term changes (that are likely to reflect practice-related improvements) or short-term fluctuations
are general across tasks, and thereby mimic the positive manifold of between-person differences. In other words, individuals with a greater hint of $g$ in the structure of their daily fluctuations were more similar to the between-person structure than individuals with no such hint. The average loadings of the tasks on the normalized first eigenvector (with a theoretical maximum of three for nine exactly equal loadings, whereby lower values indicated less equal loadings or even some negative loadings) were 2.93 ($SE = 0.0044$) for the between-person, 2.08 ($SE = 0.13$) for the raw within-person, and 1.06 ($SE = 0.14$) for the de-trended within-person structures, indicating that the $g$ factor was less dominant for the within-person structures, particularly when practice-related trends were taken out. When comparing the within-person structures with the between-person structures at posttest, which were significantly different from the between-person structures at pretest (average KL divergence $= 4.15$; $p < .001$), the resulting average divergences were even larger (average KL divergence $= 9.77$; $p < .001$, for within-person structures based on raw data; average KL divergence $= 14.08$; $p < .001$, for within-person structures based on de-trended data). It therefore seems unlikely that the differences of the within-person structures from the between-person structures at pretest can be explained by practice-induced changes of the psychometric properties of the tasks (see Method: Validity of the Tasks) that lead to the apparent shift of the between-person structures from pretest to posttest—at least for all those participants whose within-person structures did not lie in the area between the between-person structures from pretest and posttest (Fig. 1).

When KL divergences were calculated separately for each ability factor, the within- and corresponding between-person correlation patterns still differed reliably from each other, with the distance being smallest for the working memory factor, both for raw and for de-trended data (Fig. 2). Importantly, these separate distances correlate only weakly with one another across persons (correlations for raw/de-trended data: $-.02/.03$ for perceptual speed and working memory, $.44/.19$ for perceptual speed and episodic memory, and $.31/-.13$ for working memory and episodic memory; with correlations of $.19$ or higher being significant at $\alpha < .05$). This indicates that for different individuals, the overall deviation of within-person and corresponding between-person structures can be attributed to different patterns of deviations at the level of separate abilities. Put simply, some individuals showed greater deviations for tests of perceptual speed, others for tests of working memory, and still others for tests of episodic memory factors.

The observed divergences of within-person structures from one another and from between-person structures have important implications for the predictability of behavior. At the between-person level, knowing how a person performs on a particular cognitive task allows prediction, to some extent, of her/his individual performance relative to other persons' performances on other cognitive tasks. It remains an open question, however, to what degree knowledge of a person's performance level on a particular task and a particular day also allows prediction of that person's performance (relative to her or his own average) on other tasks on the same day. To answer this question, we conducted a series of regression analyses that aimed at predicting performance of each person on each task and each day with performance of the same person at the same day on the remaining eight tasks. The regression coefficients for these other tasks were based on: either (a) the

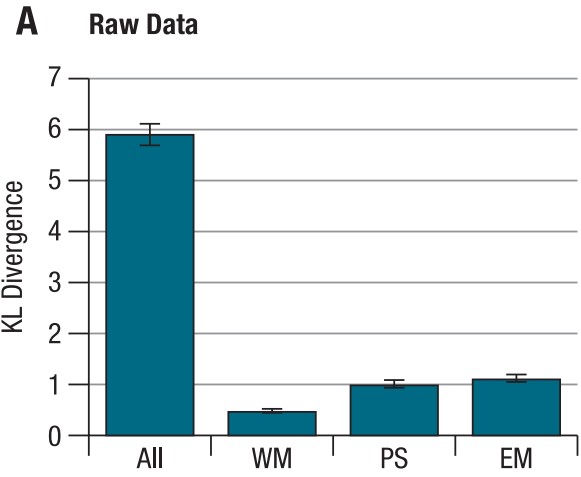

**A**     **Raw Data**

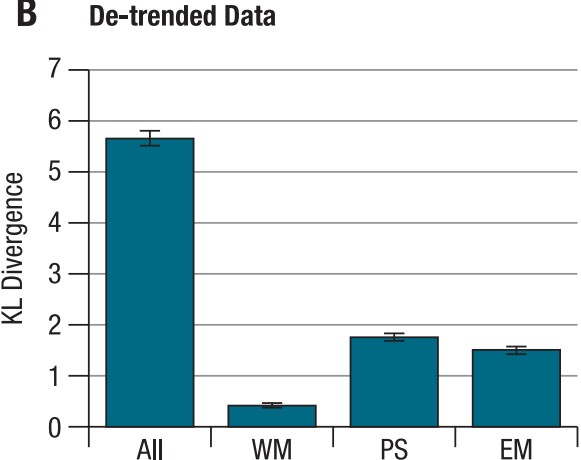

**B**     **De-trended Data**

**Figure 2**   **KL divergences between within- and between-person structures for different abilities on the basis of (A) raw, and (B) de-trended within-person data.** Calculating KL divergences separately for the different ability factors shows that within- and between-person structures differ reliably from each other for each ability. These differences are more pronounced for episodic memory and perceptual speed than for working memory. Error bars indicate the standard deviations from simulated distributions under the null hypothesis of no difference between within- and between person structures. All = all nine tasks; WM = working memory; PS = perceptual speed; EM = episodic memory.

individual within-person correlation matrix of this person, (b) the average within-person correlation matrix, or (c) the between-person correlation matrix from pretest. We ran all of these models once for the raw, and once for the de-trended data. We also conducted a set of prediction models in which we did the reverse, that is, we tried to predict between-person differences at pretest on single tasks using scores on the other eight tasks and regression equations based on information either from the corresponding between-person correlation matrix or from the individual or average within-person matrices. In total, about 90,000 prediction models (101 persons * 101 days * 9 tasks) were run. Results were averaged and
are displayed in Figs. 3A and 3B. Moreover, the results of 909 prediction models (101 persons * 9 tasks) were averaged, and are shown in Fig. 3C.

Summary results followed a consistent pattern (see Fig. 3). Predictions are best when between-person information is used to predict between-person differences and when individual within-person information is used to predict individual within-person variability. It is worst when within-person information is used to predict between-person differences and when between-person information is used to predict individual within-person variability; prediction with the average within-person structure fell in-between. It was striking to find that for almost all of the tasks, trying to predict de-trended within-person variability using between-person models did not work any better (or was even worse) than simply taking the within-person means.

We next took a closer look at the divergence of the *average* between- and within-person structures. The distribution of the correlation matrices in the MDS solution showed indications of normality in quantile–quantile plots (see Fig. S1). Therefore, the centroid correlation matrix of the within-person cluster and the centroid correlation matrix of the between-person cluster were considered as viable average representations of within-person and between-person structures, respectively. Confirmatory modelling of a hierarchical factor structure was used to compare the two average correlation matrices. The model specified first-order ability factors for episodic memory, working memory, and perceptual speed, and a second-order general ability factor. For the between-person correlation matrix, this resulted in very good model fit ($\chi^2[24] = 20.77$, $p = .998$; Root Mean Square Error of Approximation (RMSEA) < .01; Comparative Fit Index (CFI) = 1.00; Standardized Root Mean Square Residual (SRMR) = .06). Standardized factor loadings ranged from .60 to 1.00 for the perceptual speed tasks', from .52 to .84 for the episodic memory tasks', and from .46 to .50 for the working memory tasks' loading on the respective ability factors (Fig. 4A). The ability factors' loadings on the general factor were .27 for perceptual speed, .54 for episodic memory, and 1.00 for working memory (Fig. 4B).

For the centroid within-person correlation matrix of raw data, model fit was also very good ($\chi^2[24] = 9.03$, $p = 1.00$; RMSEA < .01; CFI = 1.00; SRMR = .04). However, as the number of independent observations for the average within-person correlation matrix is unknown due to possible autocorrelations of the repeated assessments, the fit indices based on $\chi^2$ (RMSEA and CFI) for this, and the analysis of de-trended data below, need to be interpreted with caution. Standardized factor loadings ranged from .71 to .78 for the perceptual speed tasks', from .46 to .53 for the episodic memory tasks', and from .54 to .65 for the working memory tasks' loading on the respective ability factors (Fig. 4A). The ability factors' loadings on the general factor were .55 for perceptual speed, .71 for episodic memory, and 1.00 for working memory (Fig. 4B).

For the centroid within-person correlation matrix of de-trended data, model fit was again very good ($\chi^2[24] = .90$, $p = 1.00$; RMSEA < .01; CFI = 1.00; SRMR = .02). Standardized factor loadings ranged from .44 to .63 for the perceptual speed tasks', from .31 to .46 for the episodic memory tasks', and from .16 to .44 for the working memory tasks' loading on the respective ability factors (Fig. 4A). The ability factors' loadings on the general factor were -.06 for perceptual speed, .82 for episodic memory, and 1.00 for

**A**    **Predicting Raw Within-Person Data**

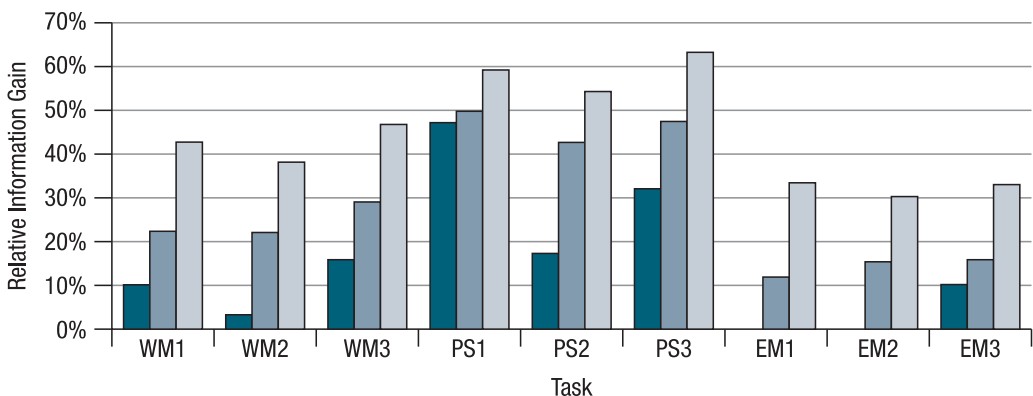

**B**    **Predicting De-trended Within-Person Data**

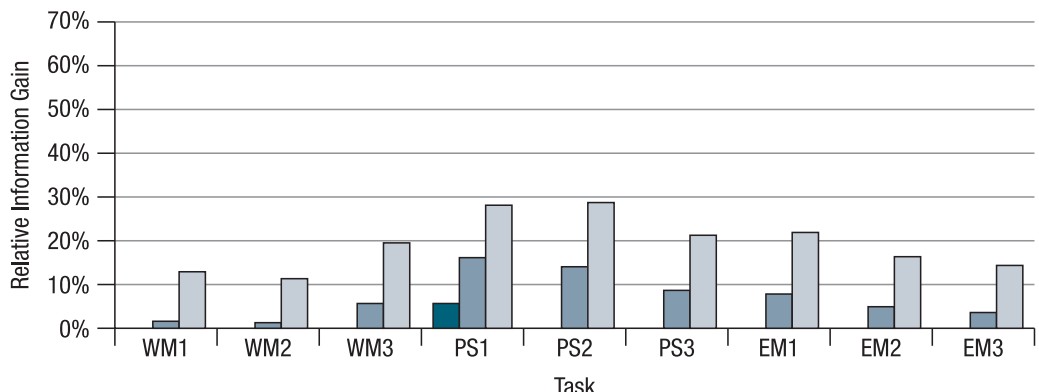

**C**    **Predicting Between-Person Data**

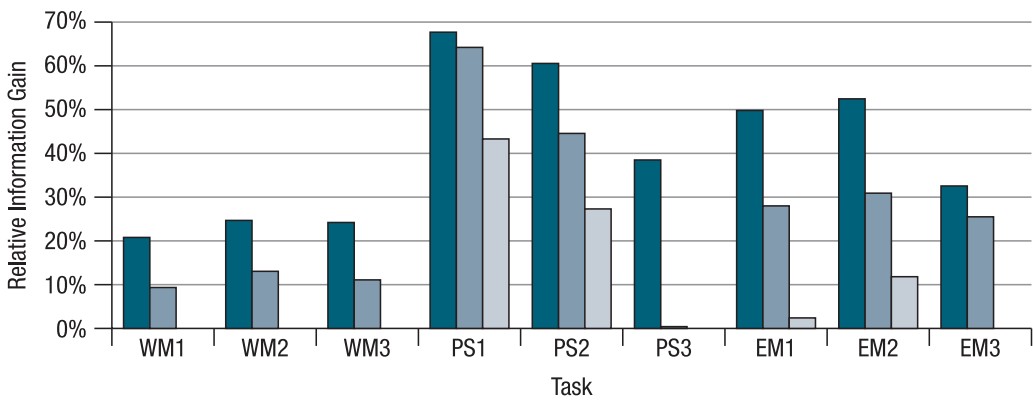

**Figure 3** **Differential predictive validity of within- and between-person structures.** Performance on each of the nine tasks was predicted by performance on the remaining eight tasks. Regression coefficients were based on between-person correlations (dark bars), average within-person correlations (middle blue bars), or individual within-person correlations (light bars). 

**Figure 3 (…continued)**
The bars show relative positive information gain compared to predicting performance with the corresponding means. Positive values can be interpreted as coefficients of determination (multiple $R^2$), while zero values refer to predictions equal or worse than prediction with the mean. (A–B) Performance of each person on each single task on each daily session (WM1–3 = working memory tasks; PS1–3 = perceptual speed tasks; EM1–3 = episodic memory tasks) was predicted by this person's performance on the other eight tasks on the respective same day. Results are shown for raw (A) and de-trended (B) within-person data. (C) Performance of each person on each task at pretest was predicted by this person's performance on the remaining eight tasks on that occasion. Predictions are best when between-person information is used to predict between-person differences (C), and when individual within-person information is used to predict individual within-person variability (A & B).

working memory (Fig. 4B). In other words, while there were only very small amounts of shared variance among the working memory tasks, that common variance was strongly shared with the episodic memory tasks once variance due to longer-term trends was taken out.

To summarize, average between-person data and average within-person raw data showed similar factor loadings for perceptual speed and working memory; for episodic memory, within-person raw data showed lower loadings than between-person data (Fig. 4A). When de-trending the data, within-person factor loadings were further reduced, particularly for the working memory tasks, indicating that shared within-person variance among tasks was to some degree due to longer-term trends (e.g., practice-related improvements). Comparing the loadings of ability factors on the general factor (Fig. 4B) revealed that the general factor was identical to the working memory factor both between and within individuals, whereas the loading of perceptual speed on the general factor was much less strong for the raw, and absent for the de-trended within-person data.

## DISCUSSION

Our results demonstrate that well-established between-person findings provide little information about correlations among day-to-day fluctuations in cognitive performance within healthy younger adults. Knowing that a given person shows high or low levels of performance on a particular task or ability relative to herself/himself on a particular day does not allow us to predict this person's performance on different tasks or abilities on the same day, unless his/her within-person structure has been assessed. Individuals showed idiosyncratic correlational patterns, resulting in weak average loadings of tasks on ability factors for de-trended data, and in ability-specific deviations of within-person structures from between-person structures. The *g* factor was less prominent within than between persons, and within-person structures with larger first eigenvalues were more similar to between-person structures than within-person structures with smaller first eigenvalues.

Whatever the mechanisms producing the larger eigenvalues at the between-person level are, these mechanisms are apparently less influential at the within-person level. One possible starting point for trying to explain the difference of between- and within-person structures therefore is to list influences that may be responsible for the positive manifold underlying the g factor at the between-person level and to consider which of these may be less (or not at all) affecting day-to-day variations in cognitive performance.

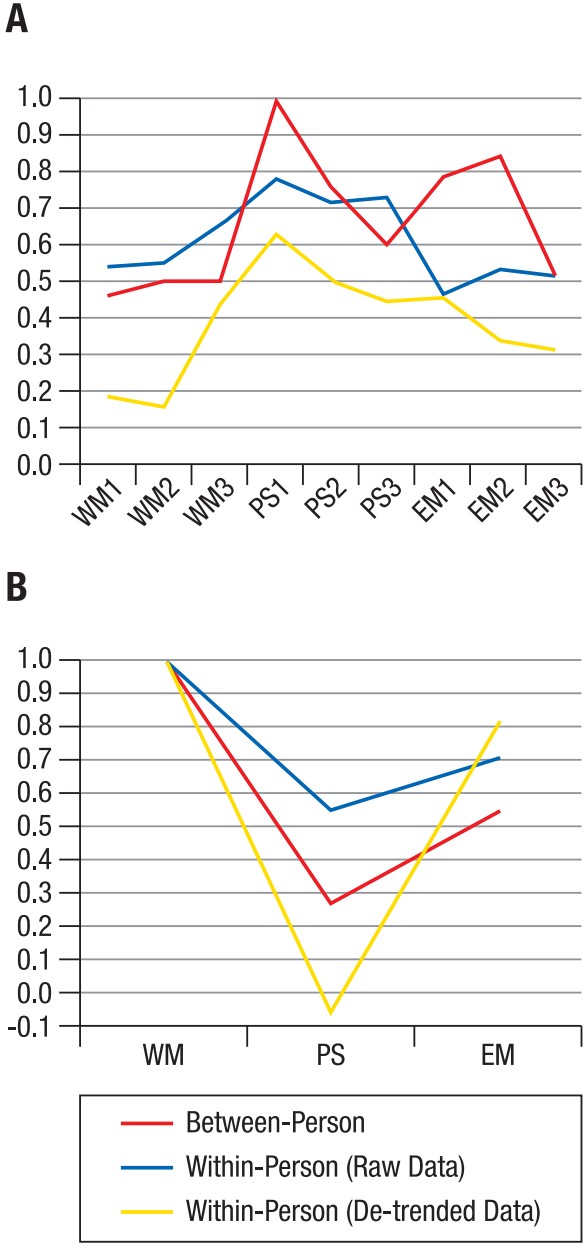

**Figure 4   Factor loadings of hierarchical models.** Factor loadings of working memory (WM), perceptual speed (PS), and episodic memory (EM) tasks on corresponding ability factors (A) and of ability factors on the general factor g (B), based on a hierarchical model applied to the centroids (average correlation matrices) of the individual structures shown in Fig. 1. At both the between-person and within-person level, the g factor was identical to WM, but the PS factor related to the g factor only when between-person or raw data within-person variance was analyzed (B).

A first way to try to explain the presence of a g factor is that one ability has a superordinate role and (partly) influences performance on the other abilities. In psychometric research on between-person differences in intelligence, such a role has been ascribed to processing speed as well as to working memory. Regarding processing speed—which can be measured
with different kinds of elementary cognitive tasks, including the kind of perceptual speed task used in our investigation—there is a long history of explaining higher-order cognition (intelligence) in terms of the speed with which necessary component processes can be conducted (*Jensen, 1998*). In our between-person models, the loading of the perceptual speed factor on a general factor was relatively low, though, and for the within-person structures of raw data, the loading of the perceptual speed factor on g was even higher than for the between-person data. At least for the selection and operationalization of abilities and tasks in our study, the role of speed in the positive manifold of correlations therefore does not serve as a prime candidate for explaining the differences of between-person and within-person structures. This is different for working memory, which had a standardized loading of 1.00 on the g factor, that is, could be equated with g, at the between-person as well as at the within-person level. This matches a combination of theoretical accounts that (a) identify g with the ability of fluid intelligence (e.g., *Gustafsson, 1984*) and (b) view working memory as the main determinant of fluid intelligence (e.g., *Conway, Kane & Engle, 2003*). While it is possible that day-to-day variations in working memory capacity constrain performance levels not only on working memory tasks, but also, for example, on the episodic memory tasks, this influence does not seem consistent enough across tasks and participants, however, to produce a more homogeneous pattern of within-person structures. A possible reason might be that the common influence of working memory, which could be identified for the average within-person structure, is covered up by idiosyncratic and time-varying patterns of task-specific strategies. Targeted analyses of single tasks of our data set indicate that such between-person and within-person adaptations of strategies play a substantial role (for the numerical episodic memory task, see *Hertzog et al., 2017*; for the spatial episodic memory task, see *Noack et al., 2013*; for the numerical working memory task, see *Shing et al., 2012*).

Differences in cognitive functioning can be assumed to reflect short-term variability, long-term change, and stable individual differences (*Hertzog, 1985*; *Nesselroade, 1991*; *Voelkle et al., 2018*). Given that the relative importance of these contributions is likely to differ by age and observation interval, the present results may not generalize to other age periods and observation intervals. Rather, the present analyses and findings are a first and admittedly descriptive step towards the general objective of delineating the driving forces of individual differences in development (*Baltes, Reese & Nesselroade, 1988*). In this vein, a second line of approaches is seeking to explain the g factor by adopting a more dynamic perspective that views the positive manifold as the outcome of long-term developmental processes. For example, *Van der Maas et al. (2006)* suggest that dynamic network models of mutual positive influences among different cognitive processes can give rise to positive manifold and a g factor, without requiring the identification of g with some common causal entity. Generally, from a developmental perspective, the presence of a pattern of between-person correlations among cognitive tasks measured in a population at a certain age has to be seen as the result of genetic endowments, environmental factors, and complex correlations and interactions among these—each of which can contribute to commonality, as well as to specificity, of performance on the different cognitive tasks (*Baltes, Lindenberger & Staudinger, 2006*; *Tucker-Drob, 2017*). In the long term, many of these processes can be

seen as cumulative within relatively stable environmental conditions. Nurturing home and education environments may therefore contribute to the development of correlation patterns of relatively stable ability trait levels that can be described by hierarchical between-person structures. Positive environments may generally foster the development of different cognitive abilities more than less optimal environments (thereby contributing to a g factor), but also support the shaping of certain ability profiles in reaction to certain genetically determined predispositions (thereby contributing to specific factors). At faster time scales, the interactions among trait-like characteristics, task-specific skills and strategies, and self-regulatory mechanisms may all be dynamically changing across time. For example, individuals may optimize task performance on one task at the cost of performance on another task on some days, and vice versa on other days, leading to transient negative correlations at the within-person level, while contributing to a positive between-person correlation due to cumulative learning on both tasks in the long run.

A third line of reasoning regarding the differences of between-person and within-person structures is based on the distinction of maximum versus typical performance (*Cronbach, 1960*). Our participants were doing all daily testing sessions in the same laboratory environments in which they also performed during the pretest and posttest assessments used to determine the between-person structures, and were always instructed in the same way to show high levels of performance, thereby providing comparable conditions supporting the exhibition of maximum performance. Yet, on a continuum from cognitive performance in high-stakes standardized testing (i.e., maximum performance) to cognitive performance on everyday tasks in everyday life (i.e., typical performance), we consider it likely that the 100 daily sessions were shifted more towards typical performance than the pretest and posttest assessments. Doing the same tasks repeatedly day by day makes them become part of daily routines that are potentially more strongly influenced by the motivational and emotional states that participants bring to the lab. Confronted with a whole battery of cognitive tasks day by day, individual patterns of task preferences may play an important role and interact with affective and motivational factors. For example, individuals may generally invest more effort on days on which they are in a better mood. On stressful days with more negative affect, however, they might preferentially try to keep up performance on a subset of tasks they like most, while sacrificing effort on tasks they enjoy less. Given that we are lacking information on mood and motivation at the level of individual tasks, these speculations cannot be empirically corroborated in the present study.

To sum up, highly complex and individualized models of within-person processes may be necessary to explain the heterogeneous within-person structures that result from making a set of cognitive tasks part of participants' daily routines during about half a year of their lives. In this context, it is important to realize that a much longer developmental history of having been confronted with a much larger variety of cognitive tasks precedes the between-person structures that can be observed at a certain point in time. Consequently, within-person structures are a more direct reflection of short-term within-person processes, while between-person structures can be seen as their indirect cumulative outcome. Attempting to better understand the factors that contribute to the differences between these levels of

analysis and aiming at identifying areas of correspondence is at the heart of the concept of *conditional ergodicity* (*Voelkle et al., 2014*). Conditional ergodicity can be achieved if time-invariant and time-varying factors that differentially affect the different levels are controlled for, so that the residual structures fulfill the necessary conditions of ergodicity (i.e., homogeneity and stationarity). The methodology to explore and approach conditional ergodicity was developed for the general case in which between-person structures are occasion-specific slices from the same data cube as the person-specific within-person structures. In our present investigation, occasion-specific slices taken from the data cube of daily testing sessions are not representing the between-person structure well, however, as interindividual differences are compressed by the individualized patterns of presentation times. As we use the between-person structures (with individually differing patterns of presentation times) assessed separately at pretest and posttest as a reference, direct tests of conditional ergodicity are not possible with the present data. Nevertheless, exploring factors that reduce the observed divergences of between-person and within-person structures when controlled for at one or the other level (or both levels) is an important goal for future analyses. This can be further aided by the recently proposed exploratory method of *ergodic subspace analysis* (*von Oertzen, Schmiedek & Voelkle, 2020*), which allows to decompose the within and between correlation matrices into subspaces that pertain uniquely to the within-person level, uniquely to the between-person level, or to both (i.e., are ergodic).

As an unprecedented empirical endeavor, our study comes with several strengths and limitations. Strengths include the mere dimensions of the study (100 complete testing sessions from 101 participants) and the fact that the chosen tasks had good psychometric properties at the between-person as well as at the within-person level (i.e., did allow to identify reliable day-to-day variations in performance). Limitations include the restriction to a certain subset of abilities operationalized each with a certain subset of tasks. Clearly, future investigations with other cognitive abilities and different task paradigms are desirable. Furthermore, the changes in validity of the tasks from pretest to posttest complicate interpretation, but likely will be difficult to be prevented by choosing different tasks. Changing what is measured by repeatedly measuring it is not restricted to the quantum world. The present data set contains a number of additional time-invariant (e.g., personality traits, genetic information) and time-varying (e.g., daily affect, day of week) variables and therefore allows for the exploration of possible factors that might contribute to the heterogeneity of within-person structures and their divergence from between-person structures. We invite interested researchers to submit their ideas and, if possible, test them with the available data set (for details, see https://www.mpib-berlin.mpg.de/research/research-centers/lip/projects/formal-methods/ctm/cogito).

As a general outline for future investigations, we think that the following directions are important to pursue: (a) further investigations of within-person structures of cognitive performance including different tasks and other cognitive abilities, (b) an exploration of the role that individual patterns of motivation, interests, task preferences, and task-specific strategies play in generating heterogeneity among within-person structures, and (c) attempts to control for such factors as well as other influential factors at the within-person and between-person level to achieve conditional ergodicity.

Investigating within-person structures of cognitive performance may also shed new light on the phenomena of differentiation of ability factor structures as a function of age and/or ability level (*Garrett, 1938*; *Spearman, 1927*). In line with our findings reported here, extant findings of between-person factor loadings and factor correlations interacting with age, ability level, or both (e.g., *Deary & Pagliari, 1991*; *Dettermann, 1991*; *Kalveram, 1965*; *Tucker-Drob, 2009*) show that between-person structures do not apply equally well to all individuals. Rather, understanding between-person differences in between-person structures may be aided by investigating between-person differences in the corresponding within-person structures. For example, it would be a fascinating research direction to study whether the progression from less differentiated to more differentiated ability structures across childhood is mirrored by an age-related trend of increasingly differentiated within-person structures.

From an applied perspective, the investigation of individual within-person structures and covarying factors opens up an entirely new field of investigation (*Boker, Molenaar & Nesselroade, 2009*; *Molenaar, Huizenga & Nesselroade, 2003*). This includes the use of individual within-person structures to dynamically predict individual cognitive performance and to attempt to improve performance by optimizing conditions and selecting and tailoring personalized interventions. Findings showing that positive between-person correlations among tasks do not ensure that practice-related performance improvements on one task lead to transfer effects on the other task (*Rode et al., 2014*) hint at the potential of within-person structures to allow for better predictions of correlated training and transfer effects at the individual level (*Lindenberger & Lövdén, 2019*).

## CONCLUSIONS

The present findings do not militate against the practical utility of hierarchical between-person structures for prediction and selection purposes in, for example, clinical, educational, or organizational psychology. However, the data show that between-person differences cannot be taken as a surrogate for within-person structures. If the aim is to describe, explain, and modify cognitive structures at the individual level, we need to measure and follow individuals over time. To understand the cognitive, motivational, and experiential mechanisms generating heterogeneity among within-person structures, researchers need to measure individual people intensively in time (*Voelkle, 2015*). Our findings indicate that the hierarchical model of intelligence is not necessarily the best template for capturing the organization of intelligence within individuals. In line with calls for person-oriented medicine (*Schork, 2015*) and person-oriented neuroscience (*Finn et al., 2015*; *Mechelli et al., 2002*), there is an urgent need for the person-oriented study of behavior (*Molenaar & Campbell, 2009*; *Nesselroade & Schmidt McCollam, 2000*). To make fundamental progress in understanding the development and organization of intelligence, we need to exploit the insight gained from following individuals over time and measure them sufficiently often to reveal the structural dynamics of their behavioral repertoire.

## ACKNOWLEDGEMENTS

We thank Julia Delius, Ray Dolan, and Manuel Voelkle for valuable comments.

### Funding

The COGITO Study was supported by the Max Planck Society, including a grant from the Innovation Fund of the Max Planck Society (M.FE.A.BILD0005); the Sofja Kovalevskaja Award of the Alexander von Humboldt Foundation (to Martin Lövdén) donated by the German Federal Ministry for Education and Research (BMBF); the German Research Foundation (DFG; KFG 163); and the German Federal Ministry for Education and Research (BMBF; CAI). Ulman Lindenberger was supported by the DFG's Gottfried Wilhelm Leibniz Award. The funders had no role in study design, data collection and analysis, decision to publish, or preparation of the manuscript.

### Grant Disclosures

The following grant information was disclosed by the authors:
Max Planck Society.
Innovation Fund of the Max Planck Society (M.FE.A.BILD0005).
The Sofja Kovalevskaja Award of the Alexander von Humboldt Foundation.
German Federal Ministry for Education and Research (BMBF).
German Research Foundation (DFG; KFG 163).
German Federal Ministry for Education and Research (BMBF; CAI).
DFG's Gottfried Wilhelm Leibniz Award.

### Competing Interests

The authors declare there are no competing interests.

### Author Contributions

- Florian Schmiedek conceived and designed the experiments, performed the experiments, analyzed the data, prepared figures and/or tables, authored or reviewed drafts of the paper, and approved the final draft.
- Martin Lövdén and Ulman Lindenberger conceived and designed the experiments, authored or reviewed drafts of the paper, and approved the final draft.
- Timo von Oertzen analyzed the data, prepared figures and/or tables, authored or reviewed drafts of the paper, and approved the final draft.

### Human Ethics

The following information was supplied relating to ethical approvals (i.e., approving body and any reference numbers):

The Ethical Review Board of the Max Planck Institute for Human Development, Berlin, granted Ethical approval to carry out the study.

## Data Availability

For reasons of privacy, the present data can only be shared with research institutions that meet the data protection requirements prescribed by European and German law. We have set up a data sharing procedure, and invite interested researchers to request the data for reanalyses from the authors. See https://www.mpib-berlin.mpg.de/research/research-centers/lip/projects/formal-methods/ctm/cogito.

## Supplemental Information

Supplemental information for this article can be found online at http://dx.doi.org/10.7717/peerj.9290#supplemental-information.

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
