# Peer review of "Within-person structures of daily cognitive performance differ from between-person structures of cognitive abilities"

_PeerJ, doi:10.7717/peerj.9290_

## Round 0.1 · original submission · Major Revisions

Thank you for submitting this revised version of your manuscript. The two reviewers have made positive comments, but some of the fundamental challenges from the review of the initial version do not appear to me to have been entirely resolved. To be honest, while there have been some substantial revisions in this version of your manuscript, I was surprised that there were not even more changes from the original version in response to the review comments (as noted then, I felt that in order to address all of these “it would become a rather different manuscript”). Having said that, you have not yet had the opportunity to present counter-arguments to some of those initial challenges.

Given the changes you have made, and the very positive reviewer comments, I will invite you to respond to these reviewers’ few comments and also to address the below comments from myself from the initial submission. The second reviewer makes important points about the implications of the findings that still need to be addressed.

I suggest thinking carefully about the title which appears to me to be claiming a null hypothesis—and which does not match with the first two sentences of the Conclusions on Lines 610–613).

I am still not aware of any argument as to why between- and within-person structures could be the same except by coincidence (as you have added to the introduction: “In technical terms, within-person and between person structures can be expected to be non-ergodic”). I continue to feel that the statistical approach being used here does not address the underlying question of whether or not the observed, and, I think we are in agreement here, inevitable, differences are important. Null hypothesis significance testing is fine for showing that there is evidence to support some degree of difference, but the question of whether the differences are of clinical relevance (can between-person differences be used as a proxy or surrogate for within-person differences—we have every reason to expect that these are not the same) is much more important and the interpretation of observed differences depends crucially on the precision of those estimated differences. In sufficiently large samples, most sensible, and plenty of non-sensible, research questions will lead to rejecting the null hypothesis (I’m confident that very large samples would find evidence of a difference in cognitive performance structures between left and right handed individuals, for example, although I have no idea which direction the effect would take). As you note in some new text: “the degree of non-correspondence of between- and within-person structures is required to lay the foundation for further research on the (potential manifold of) mechanisms that explain such non correspondence.” (Lines 100–103) and I agree that it is the degree of non-correspondence that must be of interest here, and would add that the precision of the estimated non-correspondence is also crucial for this to be interpreted. As we are all aware, there has been a strong movement away from p-values and towards considering effect sizes and precision alongside (or for some, although as a biostatistician I disagree here, instead of) this. As I noted last time: “The lack of a sample size calculation to justify the number of participants and measurement occasions raises the question of whether or not you are able to provide sufficiently precise estimates/power to detect differences of a given [meaningful] size. The research question here appears to me to be better suited to an equivalence analysis (which would require specifying a margin that can be interpreted as consistent with ‘equivalence’ and this would then have informed the sample size needed). I don’t think that statistical significance testing is the right approach to address your research question here. … The title [which has not changed from the original version] appears to be claiming a null (rather than a failure to reject a null) hypothesis (‘Within-person structures of daily cognitive performance cannot be inferred from between-person structures of cognitive abilities’), whereas the penultimate sentence in the abstract [which also seems to be unchanged from the previous version] suggests a lack of equivalence (‘We conclude that between-person structures of cognitive abilities cannot serve as a surrogate for within-person structures.’)” The adequacy of your sample size would be reflected in the precision of estimates and while your claims on Lines 593–594 may well be true, the sample size in and of itself is not a strength; high levels of precision would be a strength though, irrespective of the sample size used to achieve this. I appreciate the addition of two examples to illustrate the magnitude of the differences (“The resulting KL divergence of these two models would be 0.80. The observed divergences, on average, therefore were considerably larger than this theoretically already quite meaningful difference.” and “To further illustrate a KL divergence of 5.90, we picked the participant with an individual KL divergence (of 5.95) closest to this average value.” (Lines 348–352) Supplementary table 1 is useful in showing that the point estimates do indeed vary between this “prototypical” participant and the sample as a whole pre-test but this still omits the variability of the estimated divergence(s). Other participants with similar KL divergences could easily have very different patterns of departures from the between-participant structure. You have clearly established that there is evidence supporting differences in within- and between-person structures, but this was entirely expected and while confirming that which is expected is still of value (as noted by both reviewers), the magnitudes of differences and their precisions seem to me to be more important here.


I greatly appreciate the additions of information about participants’ demographics and some indication of study attrition. I was still unsure of how participants were recruited (including inclusion and exclusion criteria) from this manuscript alone.


I feel that the discussion could be extended by looking more closely at the implications of your findings for researchers and for practitioners (see comments from Reviewer #2) which would help establish this study as part of a broader programme aiming to address the underlying questions. You do touch on this in places (e.g. Lines 597–598), but I feel that you have more to say about future steps in this area.

While I appreciate the point on Lines 178–179 about outliers, removing these values based solely on their unusualness doesn’t seem appropriate. Could you explain this part in more detail?


Finally, a reviewer of the original submission asked “We can start with the KL test. How is it that this is the best procedure to use for the study’s intended purpose?” While I appreciate that you say “The KL divergence is an appropriate metric for this question because it provides a symmetrical measure of how much information (measured in nats = 1.44 bits) is lost when one statistical distribution (i.e., a between person correlation matrix) is used to describe another distribution (i.e., a within-person correlation matrix; see below for further information).” (Lines 114–118) I don’t think this establishes for the reader why a metric for comparing probability distributions is appropriate in this case. To echo this reviewer, why do you believe that KL is the best metric here?

Reviewer 1 ·

Basic reporting

no comment

Experimental design

no comment

Validity of the findings

no comment

Additional comments

I reviewed this paper in March and as I mentioned then, the paper is well-written, the data set is unique, and the statistical analyses are well-chosen and sound. The authors addressed all points of my review. I appreciate their careful work and responsiveness to all the feedback they received. I have no further comments on the manuscript.

I was astonished to read the first decision to reject this paper. When reading the decision and the other reviews, I had the impression that the most important disagreement was about the potential contribution of the paper to the field. I understand that by mere logic, between- and within-person structures should not be the same and I see why this would not be surprising for some readers. However, the implications of this at first glance intuitive finding are far reaching. It implies that as a field, we must change the methods in which we used to study the organization of cognitive abilities. Even in 2019, our typical designs assess almost always between-person structures and thus do not inform us (not even approximately) about individual cognitive performance. Although mere logic or, for example, the work of Peter Molenaar (2004) could have changed the way we study cognitive abilities, they obviously didn’t. In my view, an empirical investigation with such a large and unique data set (101 adults performing nine cognitive tasks on 100 occasions) could be just the empirical evidence we need as a field to no longer be able to ignore this issue. Thus, although the main finding could be considered intuitive, I expect it to have a major impact.

Molenaar, P. C. M. (2004). A manifesto on psychology as idiographic science: Bringing the person back into scientific psychology, this time forever. Measurement, 2, 201–218.

·

Basic reporting

Pass

Experimental design

pass

Validity of the findings

pass

Additional comments

I was the most positive reviewer from the first submission, and I still am positive about this work. It is a bit strange to me that the prior reaction was "obviously the within and between person correlations won't be the same" and yet no one had previously systematically tried to examine why or how, and the typical language used to write about the structure of intelligence (that I have read) seems to ignore this distinction. If the general conclusion is obvious or not, the uniqueness of the data set, and the rhetorical point seem to make this worthy of publication. Though I certainly see why it is fair to point to a data analytic method problem as a reason to delay/deny publication.
And I do note again that intelligence research is not my primary area of expertise, so my support for the article may be naive....

Regardless, upon a second reading, if it's an easy enough thing to do, I would just want some discussion of further implications of the findings. For example, let's say that researchers, clinicians, and educators have been inappropriately extending their interpretation of the structure of intelligence of from between-person data to explain, predict, or treat something about the structure of intelligence within an individual? If so, what kind of thing might people be getting wrong? For example, let’s say there educational interventions geared towards improving verbal working memory for children with low scores (and let’s assume those interventions work). Further presume that research into this intervention wanted to test whether improving verbal working memory transfers to other cognitive abilities. It seems sensible that the factor structure of intelligence would predict what other abilities are most likely to also be improved, yes? Would such investigations be likely to result in type-2 errors wherein no transfer of training effects were detected because researchers selected the wrong abilities to test because they were based on a between-subjects factor structure, and not within?
Implications to discuss do not need to be applied; e.g., if the factor structure of intelligence emerges with development (e.g., abilities become more differentiated as children age), I imagine the predicted differentiation pattern would be quite different depending on using within vs. between correlations.
I realise all of this kind of discussion would be speculative, but I think some discussion along these lines could illustrate why these findings are important to consider for readers who have the reaction of “these results are obvious”

---

## Round 0.2 · Minor Revisions

Thank you yet again for your thoughtful and constructive revisions and responses. I am much more comfortable with this version of the manuscript given the new title and the additional work on interpreting the significance of differences. I appreciated the additional information about participant recruitment and further work to be done is also clearly laid out. At this point, I feel that there is one substantial issue that I still need reassuring on.

While I’m familiar with Kullback-Leibler divergence to compare probability distributions, as a biostatistician, my previous question about its appropriateness here, which was also on behalf of one of the original reviewers, was about its use for comparing correlation matrices. Perhaps this is more common than I am aware, but I have not been able to find applications of KL divergence matching yours. Are you able to provide references somewhere around Line 287 for other research taking this approach or otherwise addressing the question of why KL divergence is the right metric (I do appreciate your point about Euclidean distances and covariance matrices)? This was how I was hoping you would continue the sentence starting “The KL divergence is an appropriate metric for this question because…”.

On the basis that other readers might also wonder about this point, and I’d be happy to hear your explanations of why they are unlikely to if you feel that would the case instead, can you provide a fuller explanation/justification for this approach, ideally with references? At the moment, the only references under “Kullback-Leibler divergences” and “Statistical testing with KL divergences” are to the original article.

Related to this, in experimenting with my own simulations, I have found myself making assumptions rather than being certain about exactly what was done here. Perhaps I’m simply misunderstanding some part of the statistical methods, but if so, this might indicate the value of the program code (irrespective of the language/software used) to provide an unambiguous statement of the algorithm?

·

Basic reporting

No comment

Experimental design

No comment

Validity of the findings

No comment

Additional comments

I'm now satisfied with the paper.

---

## Round 0.3 · accepted · Accept

Thank you for your responses and also for your patience during these challenging times.

I think you have done an excellent job in revising this manuscript over its several iterations in response to comments from the reviewers and myself. I feel that this version of the manuscript, which I am delighted to accept, provides a clear story and one that I’m sure will generate interesting and useful discussion. Well done!

I have made a few minor comments below which you can incorporate into the proofing of your manuscript as you see appropriate.

Line 42: Perhaps “consensus view” (as you use on Line 26) would be more usual phrasing rather than “consensual view” here (although one of its meanings does match your intention here).

Line 57: “One example are allelic variations…” doesn’t match (singular–plural). “Examples are…” or “One example is…” would both work here.

Line 62: The comma in “may influence both, differences between” seems spurious to me.

Lines 96–98: This is currently written as an absolute statement (requiring only a single counter-example to make it incorrect). You might prefer to make this conditional (e.g. “To the best of our knowledge…”) so that if some obscure research has addressed this in that way, your statement remains true.

Line 99: Do you mean plural “individualS’” here or “in AN individual’s levels”?

Line 125: Given n=101, the decimal place here seems spurious (each person contributes 0.99%) and slightly inconsistent with percentages given later (e.g. Lines 144–146).

Line 395: Rather than “did show”, you could consider “showed” here.

Line 472: While this is perhaps stylistic, I think “< .01” would be better than “= .00” (I appreciate that the latter, the current version, can be interpreted as rounded/truncated equivalence, but the former seems clearer to me and matches your treatment of very small p-values, e.g. Lines 415, 416, and 417). The same point would also apply to Lines 480 and 489.

Lines 484–486: You haven’t used the same possessive apostrophes here as on Lines 474–475 and Lines 489–491. While they initially seemed out of place on those earlier lines, the final “loadings”, as also used here and later, justifies their use. These sets of lines seem like they should be consistently expressed.

Line 581: Should “at some days” here be “on some days”? (c.f. later on this line “on other days”)

Line 598: Perhaps “days at which” should be “days on which”.

Line 607: Depending on the emphasis you want here, “variety of” might work better than “number of” with regards to cognitive tasks.

Lines 716–717: Journal title is not italicised here.

Line 721: Book title is not italicised here.

Lines 847–848: I’m assuming that this has now been published as doi: 10.3390/jintelligence8010003?